green chemistry

deep desulfurization, carbon nanotubes, thiophene

**Author for correspondence:**
Yue Liu
e-mail: dlut_luna@126.com

This article has been edited by the Royal Society of Chemistry, including the commissioning, peer review process and editorial aspects up to the point of acceptance.

# Deep desulfurization performance of thiophene with deep eutectic solvents loaded carbon nanotube composites

## Yue Liu, Jiaojiao Xue, Xin Zhou, Yingna Cui and Jingmei Yin

Dalian University, Dalian 116622 People's Republic of China

YL, 0000-0003-3184-8592

One source of air pollution is the combustion of sulfur compounds in fuel oil. Reducing sulfur content in fuel oil has become a hot issue demanding timely solutions. Using ionic liquids and deep eutectic solvents (DESs) to remove sulfides in fuel oil has achieved good results presently. However, since DESs are liquid and their transportation and separation are inconvenient, a new way is proposed that the DESs are loaded on the carbon nanotubes (CNTs) with large specific surface area and good chemical stability. A series of composites materials (DESs/CNTs) were prepared. Finally, they are applied to the removal of sulfides in fuel oil. This loading method, which imparts introduced unique physico-chemical properties of the DESs to the carrier materials, preserves both advantages while overcoming some of the problems with DESs. The interaction between DESs and CNTs is mutual promotion. Therefore, this study has important theoretical significance and industrial application value. Under optimal conditions, when the reagent ChCl/p-TsOH (1 : 2) was loaded on multi-walled CNTs (OD = 30–60 nm) to prepare the composite material (ChCl/p-TsOH)/CNTs, the single desulfurization rate of the composite material was 95.8%. Finally, the catalytic/oxidation mechanism was studied systematically and this work would provide a green route for the desulfurization of fuels.

## 1. Introduction

Sulfur compounds exist in petroleum, which produces sulfur oxides ($SO_x$) in the process of combustion. It will lead to environmental and health concerns. As a result, governments

around the world are enforcing strict standards to produce ultra-low-sulfur fuels from the petro-chemical industry. Thus, deep desulfurization of fuels is a significant process in industries [1–3].

Currently, as a traditional technical method, hydrodesulfurization (HDS) is extensively used in the purification of diesel oil. HDS is high in energy consumption, and the efficient reduction ability of heterocyclic S-compounds such as benzothiophene (BT), dibenzothiophene (DBT) and their derivatives is limited [4]. In the past, scientists studied many different methods for desulfurization, including adsorption [5], oxidative desulfurization [6], biodesulfurization [7] and extractive desulfurization [8,9]. Extractive desulfurization (EDS) has been widely studied because of its simple operation and low cost [10]. However, organic solvents are commonly used as extractants during the desulfurization process; because of their variability and flammability, this brings again environment and human health problems. In 2001, Bösmann et al. [11] used acidic ionic liquid (IL) [BMIM] Cl/AlCl$_3$ to extract sulfur from simulated oil DBT to achieve the purpose of desulfurization. But there are also some problems in industrial use, such as the high cost of ILs, the complexity of toxic synthesis and the difficulty of purification [12]. A possible solution is to use deep eutectic solvents (DESs) as a sustainable and low-volatile alternative to traditional organic solvents. As an analogue of ILs, DESs are defined as a mixture of two or more compounds connected through hydrogen bonding, and melting point is significantly lower than any of the constituent compounds. They can be synthesized from inexpensive, non-toxic, fully biodegradable materials and are simple to prepare [13–17]. Therefore, DESs are considered an alternative to organic solvents. Applying DESs to various sides have attracted wide attention [18]. However, compared with conventional ILs technology, DESs have less application in fuel desulfurization. Thus, it is significant to study the extraction desulfurization performance of DESs. In 2007, Abbott et al. [19] investigated the use of DESs to extract residual glycerol from fuels. In 2013, for the first time, a series of DESs for the removal of organic sulfide in fuel oil was designed and synthesized. In optimal conditions, the extraction efficiency of one cycle for BT could reach 82.83% [20]. In order to improve the DESs, in the industry it is not convenient for transportation and storage according to the documents referred to in the solid immobilized ILs [21,22]; the concept is further put forward of a new method, the DESs load on the specific surface area of carbon nanotubes (CNTs) with good chemical stability, and are then applied to the sulfur removal in the process of fuels. CNTs have been researched extensively due to their unique structure, and excellent mechanical properties, as new nanomaterials, which also have larger pore diameters and specific surface areas [23]. Crespo & Yang used single-walled carbon nanotubes (SWNTs) to adsorb gaseous thiophenes (Ths) and confirmed that most of them were adsorbed in the tube. SWNTs with smaller diameters had stronger adsorption and larger sulfur capacity [24]. Goering & Burghaus used thermal desorption spectroscopy to study the kinetics of Th adsorption on SWNT and initially proved the adsorption of Th molecules on SWNT [25].

In this paper, a series of two-component DESs were prepared and employed. Three different types of CNTs were used as a supporter to prepare composite materials. Th was chosen as a typical model sulfide. The extraction desulfurization process of Th from fuel oil was investigated by using composites and different oxidants (tert-butyl alcohol, cyclohexanone peroxide hydrogen peroxide, periodic acid, hydrogen peroxide), respectively. The influence of some factors on desulfurization efficiency was analysed systematically, including the structure of DESs, the oxidant species, oxidant dose, the mass ratio of DESs loading, CNT type, amount of composite materials, temperature, time and rotation speed. Finally, the extraction mechanism was investigated using FT-IR spectra.

# 2. Experimental

## 2.1. Synthesis and characterization of DESs

Hydrogen bond acceptors (HBAs) were based on quaternary ammonium salts, and hydrogen bond donors (HBDs) were based on some typical organic acids. The synthesis of DESs was carried out in a round-bottom flask. Raw materials were purified before use. The HBA and HBD were mixed at a proper molar ratio, and the temperature was controlled at 80°C–120°C. The two substances were heated and stirred with a magnetic stirrer until they formed a homogeneous and transparent liquid, and the reaction lasted for approximately 4 h. The synthesis of DESs was illustrated by the synthesis of ChCl/p-TsOH. Choline chloride and p-toluenesulfonic acid were mixed in a 50 ml round-bottom flask with a magnetic agitator at a 1 : 2 molar ratio. After they were heated to a liquid, the system was stirred for 4 h at 80°C. The synthesis route is shown in figure 1.

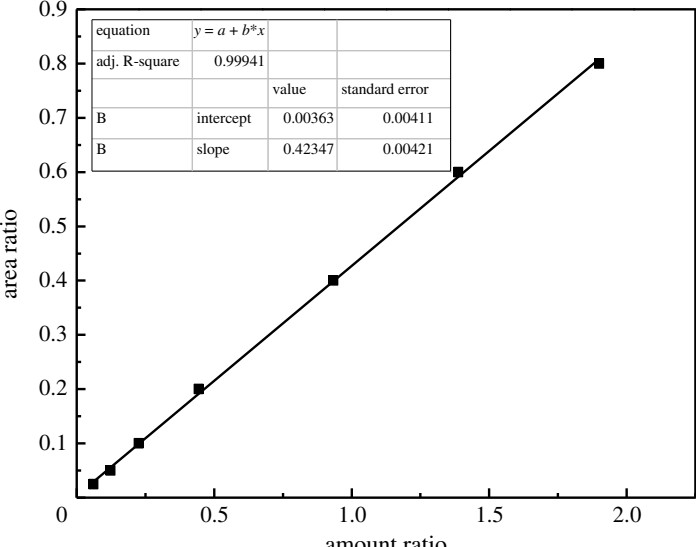

**Figure 1.** The synthesis of ChCl/*p*-TsOH.

**Figure 2.** The standard curve of Th.

## 2.2. Preparation of DESs/CNTs

A physical impregnation method was used for experiments to obtain DESs/CNTs composite materials. The synthesized DESs were loaded onto the CNTs by this method. The specific method is as follows: take 0.1 g of CNTs and a certain amount of DESs in a round-bottom flask, which are sonicated for 3 h at 65°C and magnetically stirred for 2 h in an 80°C oil bath. The mixture was taken out and then naturally dried at room temperature to obtain the DESs/CNTs composite materials.

## 2.3. Catalytic/oxidation process

Th was selected as typical sulfide and n-octane as typical fuel oil. The model oil had a sulfur concentration of 1600 ppm. The catalytic/oxidation desulfurization processes were conducted in self-made equipment. Adding DESs/CNTs, oxidant and model oil to pear-shaped bottle in a certain molar ratio. The mixture was stirred in a 25°C water bath for 60 min except for otherwise defined.

## 2.4. Analytical methods

Analysis of sulfur concentration in simulated desulfurized oil was by gas chromatography. The chromatographic conditions are shown as follows: chromatogram column: HP-5; injection volume: 0.06 µl; carrier gas (N$_2$): 210 ml min$^{-1}$; H$_2$: 40 ml min$^{-1}$; air: 350 ml min$^{-1}$; flux: 1.6 ml min$^{-1}$, constant current mode; inlet temperature: 250°C; detector temperature: 250°C; column temperature: heating from 60°C to 220°C with 30°C min$^{-1}$ increase. Th concentration was determined by the standard internal method. The n-hexadecane was chosen as the standard internal solvent and the concentration was 2000 ppm. The analysis of the standard curve is shown in figure 2. The correlation coefficient was found to be 0.99941. The values of extraction efficiency (EE) were obtained by relating the amount of

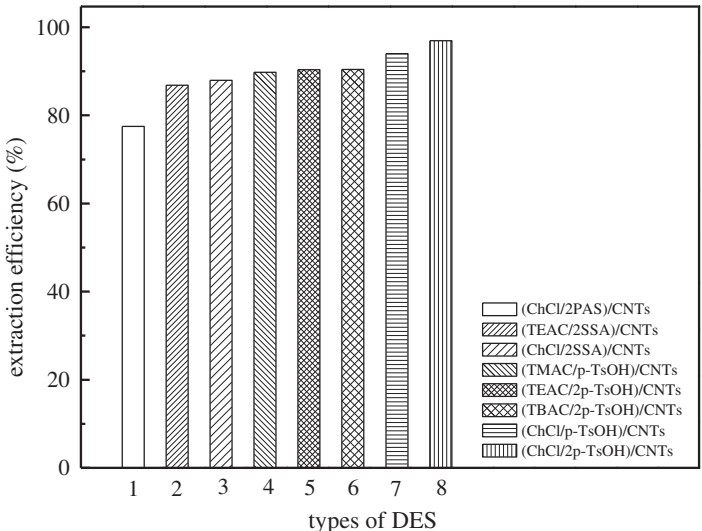

**Figure 3.** Effect of DES types.

sulfur compound in the fuel oil phase before (Ci) and after desulfurization (Cf), as shown in equation (2.1). In this study, all experiments were repeated to ensure reproducibility with an error of less than 3%.

$$EE(\%) = \frac{Ci - Cf}{Ci} \times 100. \tag{2.1}$$

# 3. Results and discussions

## 3.1. Effect of DES types

HBD and HBA of DESs play a great role in the process of desulfurization. In this work, 5-sulfosalicylic acid (SSA), p-toluenesulfonic acid (p-TsOH) and p-aminosalicylic acid (PAS) were selected as HBDs, and tetramethylammonium chloride (TMAC), tetraethylammonium chloride (TEAC), tetrabutylammonium chloride (TBAC) and choline chloride (ChCl) were selected as HBAs, and a series of DESs (TEAC/ SSA, ChCl/PAS, ChCl/SSA, TEAC/p-TsOH, TBAC/p-TsOH, TMAC/p-TsOH, ChCl/p-TsOH) were designed and synthesized which were loaded onto large-aperture CNTs (OD = 30–60 nm) carriers for sulfur removal in fuel oil. As presented in figure 3, compared with HBA, HBD has a greater impact on the desulfurization process. The stronger acidity of HBD is accompanied by higher desulfurization efficiency. In the case that HBA is ChCl when the molar ratio is the same, the order of desulfurization efficiency is as follows: p-TsOH > SSA > PAS. When HBD is SSA, the desulfurization efficiency of ChCl/SSA composite is higher than that of TEAC/SSA. Also, the molar ratio of HBA to HBD affects desulfurization efficiency. When DES is ChCl/p-TsOH (HBA : HBD = 1), the desulfurization rate of the composite is 94.04%, and when DES is ChCl/2p-TsOH, the desulfurization rate is 95.8%. In the next factors exploration, mostly ChCl/p-TsOH and TEAC/SSA were used for all the experiments. ChCl/p-TsOH was chosen because it has the best desulfurization efficiency for composite materials synthesized as a DES; although TEAC/SSA has low desulfurization efficiency for composite materials synthesized as a DES, it will be more convincing to compare it with ChCl/p-TsOH in order to obtain accurate reaction conditions. Several other DESs with a large difference in desulfurization efficiency from ChCl/p-TsOH can also be selected, of which we choose any one for comparison.

## 3.2. Effect of the oxidant

The desulfurization efficiency of the composite materials loaded with TEAC/SSA and ChCl/p-TsOH with and without introducing oxidant $H_2O_2$ is shown in figure 4a. After adding $H_2O_2$, the desulfurization efficiency reached 95.8% compared with the efficiency of 90.44% without any oxidation. Therefore, the oxidant is necessary for the deep desulfurization of fuels. Oxidant species affect the desulfurization efficiency; the desulfurization effects of four types oxidants (tert-butanol

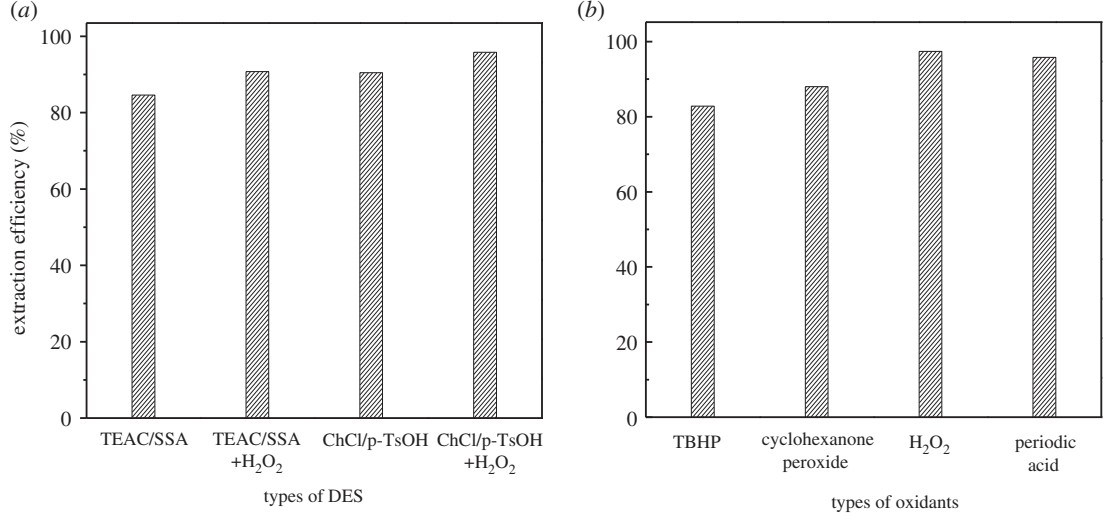

**Figure 4.** (a) The need for the introduction of oxidant and (b) effect of oxidant types with ChCl/p-TsOH as DESs.

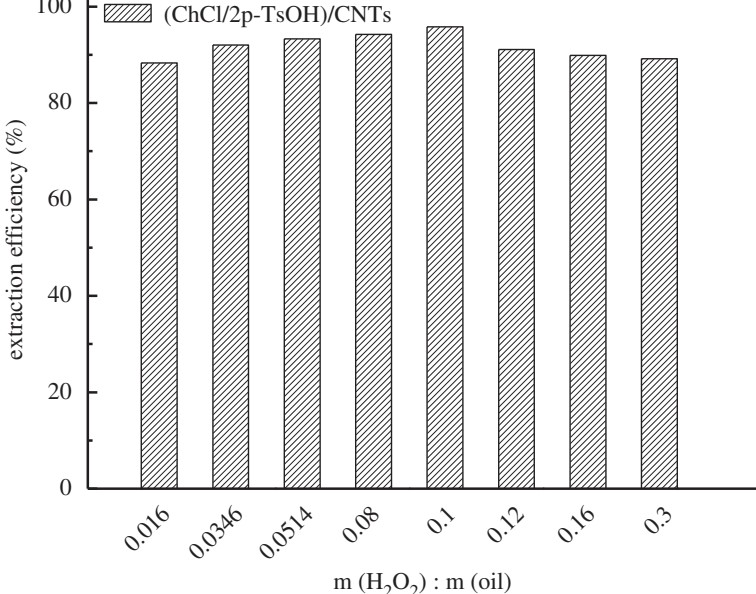

**Figure 5.** Effect of oxidant dose.

hydrogen peroxide (TBHP), cyclohexanone peroxide hydrogen peroxide, periodic acid and hydrogen peroxide) were explored. The results are shown in figure 4b; when the periodate and hydrogen peroxide were chosen as oxidant, the desulfurization efficiency was high, but considering that the oxidation product of $H_2O_2$ is water and has no impact on the environment. So $H_2O_2$ was chosen as the oxidant. Oxidant dose is a significant factor in catalytic oxidation desulfurization, and it is important that a small amount of oxidant can achieve high desulfurization efficiency. The effect of oxidizing agent on desulfurization performance was investigated. Studies show that the desulfurization efficiency gradually increases with the increase of m $(H_2O_2)$/m (oil) (figure 5). When m $(H_2O_2)$/m (oil) is 0.1, the desulfurization efficiency is 95.8%. When the mass ratio continues to increase to m $(H_2O_2)$/m (oil) is 0.3, the desulfurization efficiency decreases. Therefore, the desulfurization system selects m $(H_2O_2)$/m (oil) = 0.1 as the optimal ratio.

## 3.3. Optimization of the desulfurization process

Some factors, such as the mass ratio of DESs loading, CNT type, amount of composite materials, temperature, time and rotation speed, will affect the desulfurization process. Therefore, these factors are examined in detail.

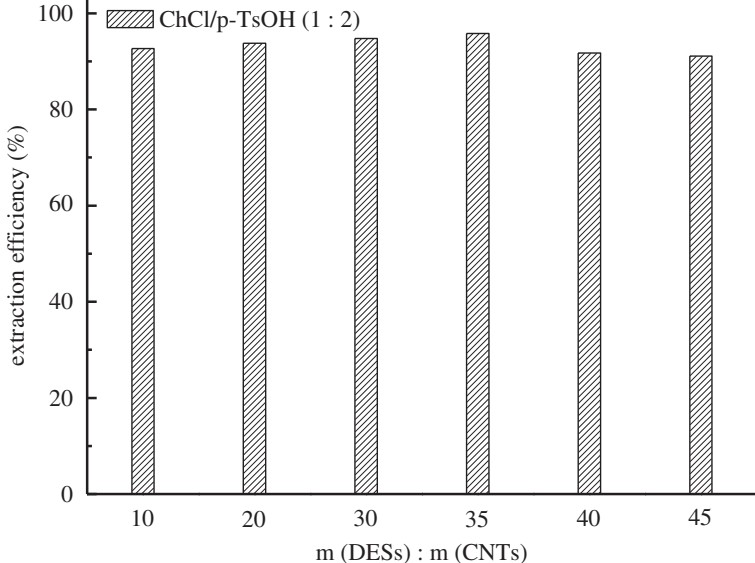

**Figure 6.** Effect of the dose of DES.

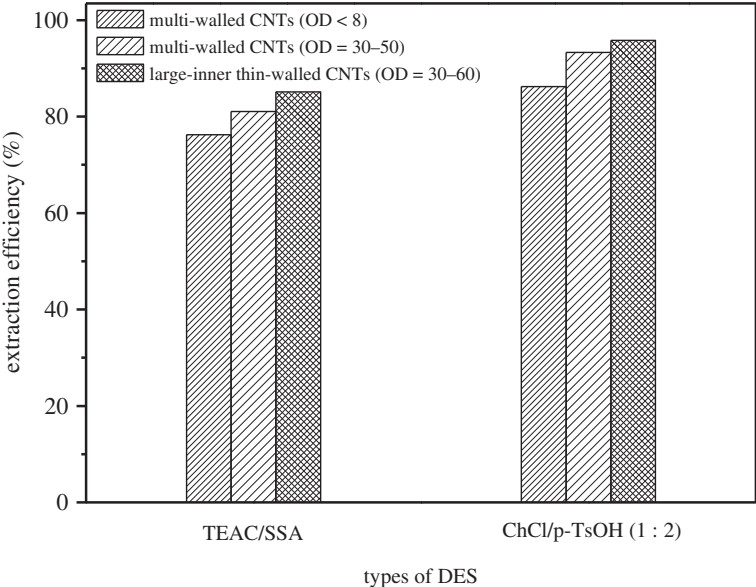

**Figure 7.** Effect of carbon nanotube types.

ChCl/p-TsOH was selected as the loading reagent to study the influence of the mass ratio of DESs on desulfurization effect, as presented in figure 6. The desulfurization efficiency increases with the increase of DESs in a certain range. When the mass of DESs load is 3.5 g with 0.1 g CNTs, the desulfurization efficiency reaches 95.8%. However, as the loading continues to increase, the desulfurization efficiency of the composite materials decreases. The main reason is that excessive DESs loading reduces the specific surface area of CNTs. The impregnated DESs gradually block the pore, thicken the pore wall, carry the pore and make it difficult for Th to enter the desulfurization process. Therefore, 3.5 g is the best DESs load.

ChCl/p-TsOH and TEAC/SSA were selected as loading reagents and loaded on three kinds of CNTs (OD = 30–50 nm, OD < 8 nm and OD = 30–60 nm) with different aperture respectively to synthesize composite materials (DESs/CNTs). The desulfurization effect is shown in figure 7. The results indicated that the desulfurization performance was the best when loaded on the large-aperture CNTs carrier (OD = 30–60 nm), and the desulfurization efficiency was 95.8% when ChCl/p-TsOH loaded on the multi-walled CNTs carrier (OD = 30–60 nm). It was found that, with the increase of the aperture of CNTs, the adsorption of multi-wall CNTs was enhanced, which was conducive to the desulfurization

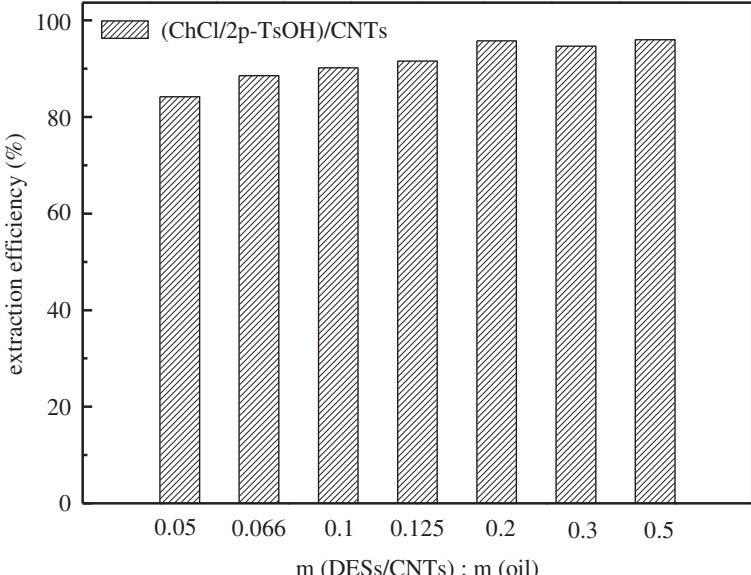

**Figure 8.** Effect of the amount of composite materials.

reaction and thus improved the desulfurization efficiency. The adsorption desulfurization efficiency of DBT by the CNTs was 60.88% in Sadare's group [26]. The CNT/TiO$_2$ nanomaterials initially afforded approximately 45% adsorption removal of DBT, 55% BT and more than 65% Th compounds from model fuels in Saleh's group [27]. Only the adsorption of CNTs has a low desulfurization rate for fuel oil. The DBT oxidation catalysed by CNTs@PDDA@Mo$_{16}$V$_2$ can reach to 99.4% in Gao's group [28]. Novel CNT hybrids with good catalytic properties can be obtained by a facile and fast method for immobilization of WO$_3$/MoO$_3$ on the surface and into the cavity of CNTs. The best oxidative desulfurization results (99.2%) have been obtained on the 2WO$_3$/MoO$_3$-CNT with the 16.5 : 8.5 wt% of WO$_3$ : MoO$_3$ in Afsharpour's group [29]. However, both of these composites with high desulfurization performance need a relatively expensive catalyst, high temperature or longer reaction time.

The effect of different amounts of composite materials (3.5DESs/CNTs) on the removal efficiency of Th was studied. It was found that the desulfurization rate rapidly increased to 95.8% when the amount of composite materials increased from 0.05 to 0.2 g. A small amount of composites contain less active components, resulting in low desulfurization efficiency. When the dosage of the composite materials was 0.2 g, a sufficient number of the composite materials can enhance the desulfurization efficiency and decrease the sulfur content of fuels, as shown in figure 8. Subsequently, the amount of composite materials was increased to 0.5 g, and the desulfurization rate remained the same. Therefore, taking considering the desulfurization rate, reaction time and cost, 0.2 g of composite materials was finally chosen as the optimal amount.

The influence of temperature on desulfurization efficiency was investigated in the range of 25°C to 60°C. As can be concluded from figure 9, the desulfurization performance of the composites increases gradually with the increase of temperature, and the desulfurization rate of the composites at each temperature can reach 92.45% when the reaction was done in 120 min, indicating that the desulfurization process is no longer affected by temperature. As the temperature increases, the equilibrium time of the reaction is shortened, and the desulfurization reaction is more likely to occur. Therefore, 25°C was chosen as the reaction temperature.

In this study, three composite materials (ChCl/p-TsOH)/CNTs, (TEAC/SSA)/CNTs and (TBAC/p-TsOH)/CNTs were used to explore the influence of time on desulfurization rate. 0.1 g of DESs/CNTs and 0.5 g of oil were selected for desulfurization reaction. When exploring the influence of the amount of H$_2$O$_2$ on the desulfurization efficiency, it was found that the desulfurization efficiency of 0.05 g of H$_2$O$_2$ was the best, so (DESs/CNTs) : oil : H$_2$O$_2$ = 1 : 5 : 0.5 was chosen. As presented in figure 10, the desulfurization rate increased with the increase of reaction time. For (TBAC/p-TsOH)/CNTs and (ChCl/p-TsOH)/CNTs, the reaction time continued to be extended for 60 min, hardly changing the desulfurization rate. The main reason is that at the beginning of the reaction, the mixed components are not fully contacted and the desulfurization rate is low. With the increase of time, the

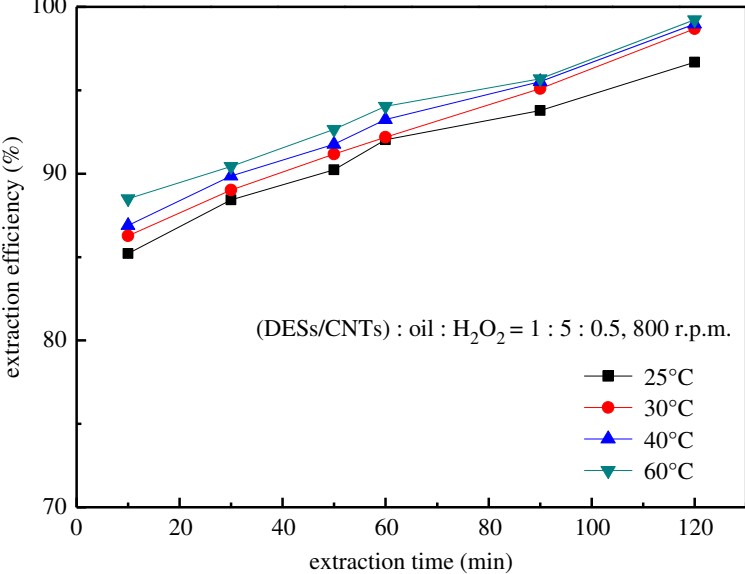

**Figure 9.** Effect of system temperature.

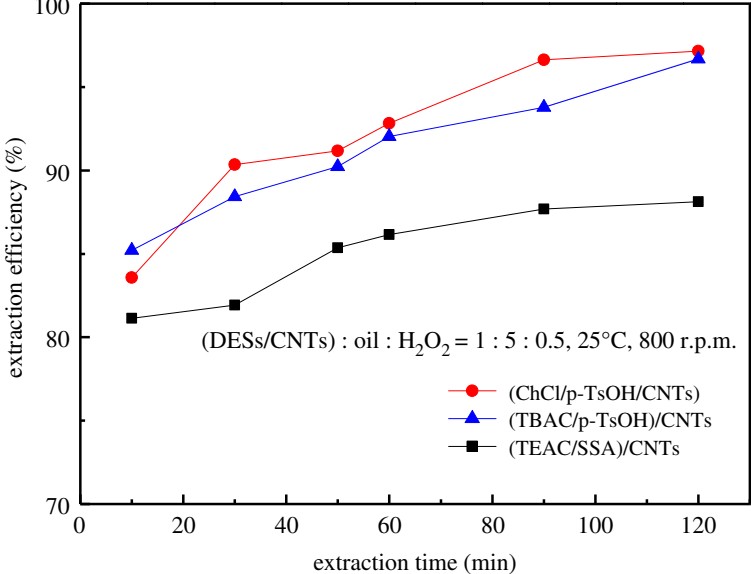

**Figure 10.** Effect of extraction time.

contact will be more sufficient so that the desulfurization rate will increase. When the reaction time is 60 min, the desulfurization rate will not change significantly with time due to the influence of too little amount of $H_2O_2$ and reaction balance. Therefore, it is appropriate to control the reaction time at 60 min.

As presented in figure 11, the desulfurization performance gradually improves with the increase of the rotational speed. The reason is that increasing the rotation speed is conducive to increasing the contact area, thereby shortening the reaction time. In order to realize the deep desulfurization, the composite material (ChCl/p-TsOH)/CNTs reached equilibrium within 60 min at 800 r.p.m., so 800 r.p.m. was chosen as the optimal rotation speed.

## 3.4. Multiple removal and recycling of composite materials

To realize deep desulfurization of fuel oil, several desulfurization processes were explored, as presented in figure 12a. After three removals, the desulfurization rate of (ChCl/p-TsOH)/CNTs was 99.29% and the sulfur concentration of fuel oil was less than 11.36 ppm. The deep desulfurization of fuel oil can be

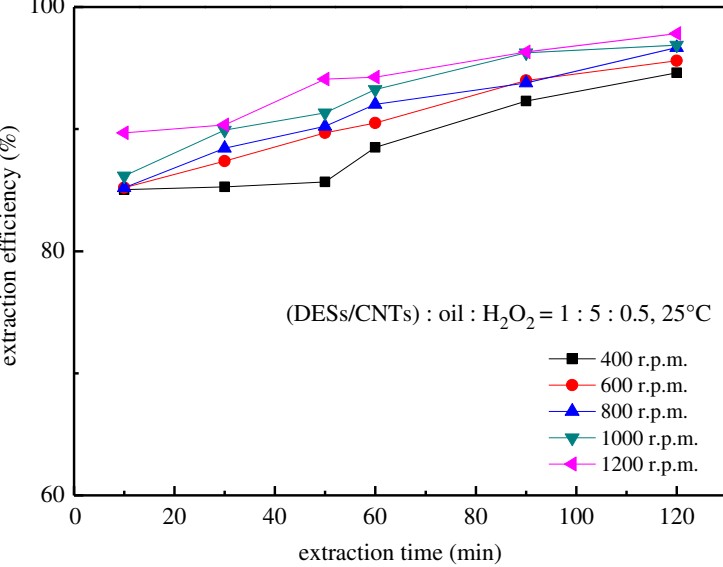

**Figure 11.** Effect of rotating speed.

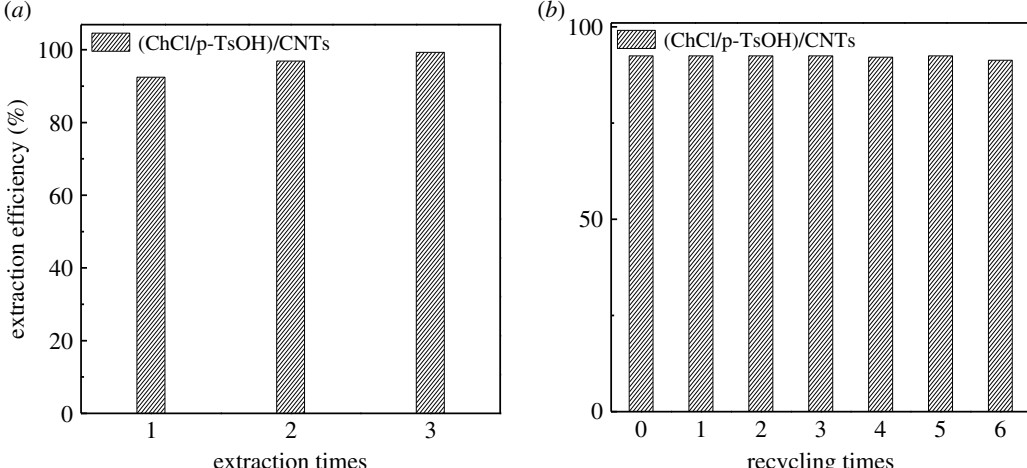

**Figure 12.** (*a*) Multiple extraction times and (*b*) effect of recycling use of composite materials.

achieved effectively, but the existing traditional techniques fail to be completed through certain cycles. It is necessary to recycle and re-use composite materials from the perspective of the economy, environment and industry. This possibility can be investigated by (ChCl/p-TsOH)/CNTs. The regeneration process was performed by dissolving the composite materials in methyl tert-butyl ether which can wash off the excess fuel oil. In the end, the remaining solution was adjusted by vacuum drying. After six recycles, the desulfurization efficiencies were quite stable (figure 12*b*). The composite materials can be recycled successfully.

## 3.5. Catalytic/oxidation mechanisms

Catalytic/oxidation mechanisms can be significant for understanding the desulfurization process thoroughly, which will also be important for the molecule design of DESs. ChCl/p-TsOH was chosen as a typical DES to explore the mechanisms. FT-IR was applied in the mechanism research, as presented in figure 13*a* and *b*. From figure 13*a*, at 705 cm$^{-1}$, Th has obvious sulfur peaks. Still, after desulfurizing by DES, CNT and DES/CNT, almost no peaks appear, indicating that they have a good desulfurization effect and can achieve the purpose of reducing sulfur concentration. As a result, during the desulfurization process, the hydrogen bond formed between the active hydrogen of ChCl/ p-TsOH and the sulfur atom of Th has a higher desulfurization rate. It can be concluded from

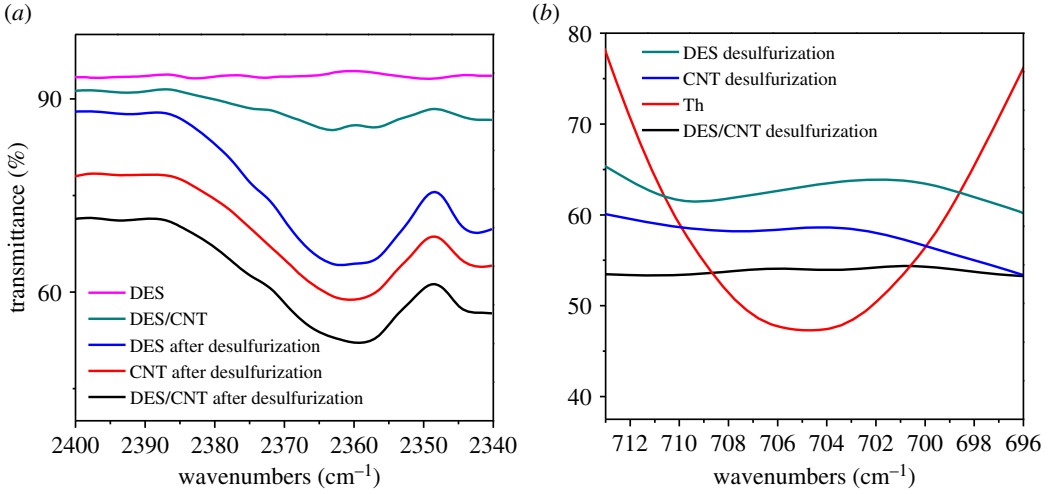

**Figure 13.** (a) FT-IR of reactive species on simulated oil desulfurization and non-desulfurization and (b) FT-IR of desulfurization and non-desulfurization of reactive species.

figure 13b that at 2360 cm$^{-1}$, after DES, CNT and DES/CNT desulfurization, there are obvious peaks, and the peak intensity of DES/CNT is greater, while DES and CNT are not. It is shown that CNT alone also has the effect of absorbing sulfur. Therefore, it can safely be concluded from FT-IR that hydrogen bonding plays an important role in the desulfurization process.

## 4. Conclusion

A series of designed DESs were synthesized and impregnated onto three different pore sizes of CNTs to prepare composite materials, which were combined with oxidants to perform the catalytic/oxidation desulfurization process of model oil. Then several suitable DESs were chosen to study the optimization of the desulfurization process. Under optimal conditions, the single desulfurization efficiency of ChCl/p-TsOH (1 : 2) can reach 95.8% by loading it on the large-aperture CNTs (OD = 30–60 nm). After three cycles, the desulfurization efficiency of the composite materials can reach 99.29%, the sulfur concentration in fuels can be decreased to 11.36 ppm or less, and finally achieve deep desulfurization. The catalytic/oxidation mechanism was researched by the numbers. Hydrogen bond was formed between DESs and Th, the DES was dispersed to CNTs with high specific surface area and a large number of microporous structures, which can provide more desulfurization activity sites, and the introduction of the CNTs make composites show certain hydrophobic properties. The sulfide in the fuel oil was fully contacted with the DES on the composite materials, and then it was removed efficiently. The research will provide a new and green way for the deep desulfurization of fuel oil.

Data accessibility. Accessible URLs: https://doi.org/10.5061/dryad.fn2z34ts5. Accession numbers RSOS-201736.

Authors' contributions. Y.L. and J.X. carried out the laboratory work, participated in data analysis, participated in the design of the study and drafted the manuscript. X.Z. carried out the mechanism analyses. Y.C. and J.Y. conceived of the study, designed the study, coordinated the study and helped draft the manuscript. All authors gave final approval for publication.

Competing interests. There are no conflicts to declare.

Funding. We would like to thank the National Natural Science Foundation of China (grant nos. 21606031 and 21473183) for the financial support of this project.

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
