## [Peer Review File · Royal Society Open Science]

Review History

RSOS-201736.R0 (Original submission)

Review form: Reviewer 1

Is the manuscript scientifically sound in its present form?

Yes

Are the interpretations and conclusions justified by the results?

Yes

Is the language acceptable?

Yes

Do you have any ethical concerns with this paper?

No

Have you any concerns about statistical analyses in this paper?

No

Recommendation?

Accept as is

Comments to the Author(s)

to many figure, maybe can combine

Review form: Reviewer 2**Is the manuscript scientifically sound in its present form?**

No

Are the interpretations and conclusions justified by the results?

Yes

Is the language acceptable?

No

Do you have any ethical concerns with this paper?

No

Have you any concerns about statistical analyses in this paper?

No

Recommendation?

Major revision is needed (please make suggestions in comments)

Comments to the Author(s)

Yue Liu et al. studied extraction of thiophene from n-octane using deep eutectic solvents loaded on carbon nanotubes as a potential green way for deep desulfurization of fuel oil. For that purpose they investigated the effects of DES type, oxidant, mass ratio of DESs loading, carbon nanotube type, amount of composite materials, temperature, time and rotation speed on extraction efficiency. Given that nanotubes have found a place for fuel desulfurization in recent years, the results of this research may contribute to this research area, but there are major revisions to be done.

Introduction section

1. Line 16: The reference 19 is not consistent with the claim. The reference is about extraction of glycerol from biodiesel. And all of the following references are inconsistent with the claims. The whole manuscript should be checked and corrected with this respect.
2. Line 37: The full name of the SWNT should be given.
3. Line 58: The factors examined should be specifically stated. The term 'and so on' does not give accurate information to the reader.

Experimental section

1. Line 11: The authors state that HBD is based on the quaternary ammonium salt or choline chloride. The fact is that choline chloride is a quaternary ammonium salt.
2. Line 45: Here 'DES/CNT' is mentioned, it should be given the full name of CNT somewhere in the manuscript.
3. In the Introduction, the authors note that composite materials were prepared in this work. However, the preparation of the composite materials is missing in the experimental part.

Results and discussion section

1. All the results are presented as extraction efficiency. It should be given the explanation and formula how it was calculated.

2. Line 53: In the Effect of DES type section, the authors explain the extraction efficiency with the DESs acidity. As the acidity of DES was not examined in this work, this way of discussing the results makes no sense. It should be corroborated with data of acidity from other sources. When authors comment the effect of molar ratio of HBA:HBD, it should be given the comparison. Here the effect of different HBA and HBD components of DES on the extraction efficiency can also be commented.
3. Line 18: In the Effect of the oxidant section and further, mostly $\text{ChCl}/p\text{-TsOH}$ and TEAC/SSA were used for all the experiments. It should be explained why these DESs were chosen.
4. Line 32: 'hydrogen peroxide' is duplicate.
5. Figure 4 - what is TBHP? It is not mentioned anywhere in the text.
6. Figure 9 - why molar ratio of (DESs/CNTs):Oil:H₂O₂=1:5:0.5 was chosen?
7. The title of section 3.4. does not correspond with the text.
8. In section 3.6., line 56, '...intensity of DES/CNT is greater, while DES and DES/CNT are not.', the second 'DES/CNT' probably should be 'CNT'.
9. Some relevant reports about using nanotubes for fuel desulfurization have been published, the authors are suggested to list their results and do some comparisons.

Reference section

1. References should be uniform as recommended by the journal.
- For the next submission, the authors are suggested to prepare the manuscript exclusively according to the instructions of the journal (to correct the text - space, italic symbols etc., graphs) and to improve the English.

Decision letter (RSOS-201736.R0)

This year has been very difficult for everyone, and we want to take the opportunity to thank you for your continued support in 2020.

The Royal Society Open Science editorial office will be closed from the evening of Friday 18 December 2020 until Monday 4 January 2021. We will not be responding during this time. If you have received a deadline within this time period, please contact us as soon as possible to allow us to extend the deadline. If you receive any automated messages during this time asking you to meet a deadline, we offer apologies and invite you to respond after the festive period or during normal working hours.

With our best for a peaceful festive period and New Year, and we look forward to working with you in 2021.

Dear Dr Liu:

Title: Deep desulfurisation performance of thiophene with deep eutectic solvents loaded carbon nanotubes composites
 Manuscript ID: RSOS-201736

The editor assigned to your manuscript has now received comments from reviewers. We would like you to revise your paper in accordance with the referee and Subject Editor suggestions which can be found below (not including confidential reports to the Editor). Please note this decision does not guarantee eventual acceptance.

Please submit your revised paper before 15-Jan-2021. Please note that the revision deadline will expire at 00.00am on this date. If we do not hear from you within this time then it will be assumed that the paper has been withdrawn. In exceptional circumstances, extensions may be possible if agreed with the Editorial Office in advance. We do not allow multiple rounds of revision so we urge you to make every effort to fully address all of the comments at this stage. If deemed necessary by the Editors, your manuscript will be sent back to one or more of the original reviewers for assessment. If the original reviewers are not available we may invite new reviewers.

On behalf of the Subject Editor Professor Anthony Stace and the Associate Editor Dr Darren Walsh.

RSC Associate Editor:
Comments to the Author:
(There are no comments.)

RSC Subject Editor:
Comments to the Author:
(There are no comments.)

Reviewers' Comments to Author:
Reviewer: 1
Comments to the Author(s)
to many figure, maybe can combine

Reviewer: 2

Comments to the Author(s)

Yue Liu et al. studied extraction of thiophene from n-octane using deep eutectic solvents loaded on carbon nanotubes as a potential green way for deep desulfurization of fuel oil. For that purpose they investigated the effects of DES type, oxidant, mass ratio of DESs loading, carbon nanotube type, amount of composite materials, temperature, time and rotation speed on extraction efficiency. Given that nanotubes have found a place for fuel desulfurization in recent years, the results of this research may contribute to this research area, but there are major revisions to be done.

Introduction section

1. Line 16: The reference 19 is not consistent with the claim. The reference is about extraction of glycerol from biodiesel. And all of the following references are inconsistent with the claims. The whole manuscript should be checked and corrected with this respect.
2. Line 37: The full name of the SWNT should be given.
3. Line 58: The factors examined should be specifically stated. The term 'and so on' does not give accurate information to the reader.

Experimental section

1. Line 11: The authors state that HBD is based on the quaternary ammonium salt or choline chloride. The fact is that choline chloride is a quaternary ammonium salt.
2. Line 45: Here 'DES/CNT' is mentioned, it should be given the full name of CNT somewhere in the manuscript.
3. In the Introduction, the authors note that composite materials were prepared in this work. However, the preparation of the composite materials is missing in the experimental part.

Results and discussion section

1. All the results are presented as extraction efficiency. It should be given the explanation and formula how it was calculated.
2. Line 53: In the Effect of DES type section, the authors explain the extraction efficiency with the DESs acidity. As the acidity of DES was not examined in this work, this way of discussing the results makes no sense. It should be corroborated with data of acidity from other sources. When authors comment the effect of molar ratio of HBA:HBD, it should be given the comparison. Here the effect of different HBA and HBD components of DES on the extraction efficiency can also be commented.
3. Line 18: In the Effect of the oxidant section and further, mostly ChCl/p-TsOH and TEAC/SSA were used for all the experiments. It should be explained why these DESs were chosen.
4. Line 32: 'hydrogen peroxide' is duplicate.
5. Figure 4 - what is TBHP? It is not mentioned anywhere in the text.
6. Figure 9 - why molar ratio of (DESs/CNTs):Oil:H₂O₂=1:5:0.5 was chosen?
7. The title of section 3.4. does not correspond with the text.
8. In section 3.6., line 56, '...intensity of DES/CNT is greater, while DES and DES/CNT are not.', the second 'DES/CNT' probably should be 'CNT'.
9. Some relevant reports about using nanotubes for fuel desulfurization have been published, the authors are suggested to list their results and do some comparisons.

Reference section

1. References should be uniform as recommended by the journal.

For the next submission, the authors are suggested to prepare the manuscript exclusively according to the instructions of the journal (to correct the text - space, italic symbols etc., graphs) and to improve the English.

Author's Response to Decision Letter for (RSOS-201736.R0)

See Appendix A.

RSOS-201736.R1 (Revision)

Review form: Reviewer 1

Is the manuscript scientifically sound in its present form?

Yes

Are the interpretations and conclusions justified by the results?

Yes

Is the language acceptable?

Yes

Do you have any ethical concerns with this paper?

No

Have you any concerns about statistical analyses in this paper?

Yes

Recommendation?

Accept as is

Comments to the Author(s)

-

Review form: Reviewer 2

Is the manuscript scientifically sound in its present form?

Yes

Are the interpretations and conclusions justified by the results?

Yes

Is the language acceptable?

No

Do you have any ethical concerns with this paper?

No

Have you any concerns about statistical analyses in this paper?

No

Recommendation?

Accept with minor revision (please list in comments)

Comments to the Author(s)

Comments to the Editor and to the Author(s) after revision

Most of the suggestions have been corrected, but the manuscript still needs minor revision. The authors are suggested to include in the manuscript their answers on Remark 6 from previous revision:

6. Figure 9 - why molar ratio of (DESs/CNTs):Oil:H₂O₂=1:5:0.5 was chosen?

Thanks for the question. In this experiment, 0.1g DESs/CNTs and 0.5g oil were selected for desulfurisation reaction. When exploring the influence of the amount of H₂O₂ on the desulfurisation efficiency, it was found that the desulfurisation efficiency of 0.05g H₂O₂ was the best, So (DESs/CNTs): Oil: H₂O₂ = 1: 5: 0.5 was chosen to study other factors.

Decision letter (RSOS-201736.R1)

Dear Dr Liu:

Title: Deep desulfurisation performance of thiophene with deep eutectic solvents loaded carbon nanotubes composites

Manuscript ID: RSOS-201736.R1

Thank you for submitting the above manuscript to Royal Society Open Science. On behalf of the Editors and the Royal Society of Chemistry, I am pleased to inform you that your manuscript will be accepted for publication in Royal Society Open Science subject to minor revision in accordance with the referee suggestions. Please find the reviewers' comments at the end of this email.

The reviewers and handling editors have recommended publication, but also suggest some minor revisions to your manuscript. Therefore, I invite you to respond to the comments and revise your manuscript.

Because the schedule for publication is very tight, it is a condition of publication that you submit the revised version of your manuscript before 10-Mar-2021. Please note that the revision deadline will expire at 00.00am on this date. If you do not think you will be able to meet this date please let me know immediately.

Kind regards,
Dr Laura Smith
Publishing Editor, Journals

On behalf of the Subject Editor Professor Anthony Stace and the Associate Editor Dr Darren Walsh.

RSC Associate Editor:

Comments to the Author:

Please do follow the suggestion by the reviewer and incorporate into your manuscript an explanation of why you chose the particular DES/CNT:oil:H₂O₂ ratio.

RSC Subject Editor:

Comments to the Author:

(There are no comments.)

Reviewer comments to Author:

Reviewer: 2

Comments to the Author(s)

Comments to the Editor and to the Author(s) after revision

Most of the suggestions have been corrected, but the manuscript still needs minor revision. The authors are suggested to include in the manuscript their answers on Remark 6 from previous revision:

6. Figure 9 – why molar ratio of (DESs/CNTs):Oil:H₂O₂=1:5:0.5 was chosen?

Thanks for the question. In this experiment, 0.1g DESs/CNTs and 0.5g oil were selected for desulfurisation reaction. When exploring the influence of the amount of H₂O₂ on the desulfurisation efficiency, it was found that the desulfurisation efficiency of 0.05g H₂O₂ was the best, So (DESs/CNTs): Oil: H₂O₂ = 1: 5: 0.5 was chosen to study other factors.

Reviewer: 1

Comments to the Author(s)

-

Author's Response to Decision Letter for (RSOS-201736.R1)

See Appendix B.

Decision letter (RSOS-201736.R2)

Dear Dr Liu:

Title: Deep desulfurisation performance of thiophene with deep eutectic solvents loaded carbon nanotubes composites

Manuscript ID: RSOS-201736.R2

It is a pleasure to accept your manuscript in its current form for publication in Royal Society Open Science. The chemistry content of Royal Society Open Science is published in collaboration with the Royal Society of Chemistry.

On behalf of the Subject Editor Professor Anthony Stace and the Associate Editor Dr Darren Walsh.

RSC Associate Editor
Comments to the Author:
(There are no comments.)

Reviewer(s)' Comments to Author:

Appendix A

Dear Dr Laura Smith,

We thank “Royal Society Open Science” for giving us the opportunity to revise our manuscript. We appreciate Dr Laura Smith and reviewers very much for your comments and suggestions on our manuscript entitled " Deep desulfurisation performance of thiophene with deep eutectic solvents loaded carbon nanotubes composites "(ID: RSOS-201736).

We have studied reviewers' comments and have revised the manuscript according to your kind advices and reviewers' detailed suggestions. Enclosed please find responses to reviewers. The original manuscript was modified based on reviewers' suggestions. Below you will find reviewers' comments (in black) and our reply to each comment (in blue).

Reviewer: 1

Comments to the Author(s)

to many figure, maybe can combine

Thanks for the reviewer's suggestion. We have combined the related figure together.

Reviewer: 2

Comments to the Author(s)

Yue Liu et al. studied extraction of thiophene from n-octane using deep eutectic solvents loaded on carbon nanotubes as a potential green way for deep desulfurization of fuel oil. For that purpose they investigated the effects of DES type, oxidant, mass ratio of DESs loading, carbon nanotube type, amount of composite materials, temperature, time and rotation speed on extraction efficiency. Given that nanotubes have found a place for fuel desulfurization in recent years, the results of this research may contribute to this research area, but there are major revisions to be done.

We would like to thank the reviewer for his constructive and insightful criticism and suggestions. We have carefully discussed all the issues raised by the reviewer as summarized below

Introduction section

1. Line 16: The reference 19 is not consistent with the claim. The reference is about extraction of glycerol from biodiesel. And all of the following references are inconsistent with the claims. The whole manuscript should be checked and corrected with this respect.

We are grateful to the reviewer for pointing out our error. After careful consideration, We have matched the references and claims in the whole manuscript correctly.

2. Line 37: The full name of the SWNT should be given.

Thanks for the reviewer's suggestion. The full name of the SWNT written is single-walled carbon nanotube.

3. Line 58: The factors examined should be specifically stated. The term ‘and so on’ does not give accurate information to the reader.

We thank the reviewer for this insightful comment. The factors are stated including the structure of deep eutectic solvents, the oxidant species, oxidant dose, the mass

ratio of DESs loading, carbon nanotube type, amount of composite materials, temperature, time, and rotation speed.

Experimental section

1. Line 11: The authors state that HBD is based on the quaternary ammonium salt or choline chloride. The fact is that choline chloride is a quaternary ammonium salt.

We are grateful to the reviewer for pointing out our error. We have corrected that HBA is based on the quaternary ammonium salt.

2. Line 45: Here 'DES/CNT' is mentioned, it should be given the full name of CNT somewhere in the manuscript.

Thanks for the reviewer's suggestion. The full name of the CNT written is carbon nanotube before DES/CNT appearing.

3. In the Introduction, the authors note that composite materials were prepared in this work. However, the preparation of the composite materials is missing in the experimental part.

We are grateful to the reviewer for pointing out our error. We have added the preparation process of composite materials in the 2.2 section. In this paper, a physical impregnation method was used for experiments to obtain DESs/CNTs composite materials. The synthesized deep eutectic solvents were loaded onto the carbon nanotubes by this method. The specific method is as follows: Take 0.1 g of carbon nanotubes and a certain amount of deep eutectic solvents in a round bottom flask, which are sonicated 3 h in 65°C and magnetically stirred for 2 h in a 80°C oil bath. The mixture was taken out and then naturally dried at room temperature to obtain the DESs/CNTs composite materials.

Results and discussion section

1. All the results are presented as extraction efficiency. It should be given the explanation and formula how it was calculated.

Thanks for the reviewer's suggestion. The values of extraction efficiency (EE) were obtained by relating the amount of sulfur compound in the fuel oil phase before (C_i) and after desulfurisation (C_f), as shown in the Eq. (1).

$$EE(\%) = \frac{C_i - C_f}{C_i} \times 100 \quad (1)$$

2. Line 53: In the Effect of DES type section, the authors explain the extraction efficiency with the DESs acidity. As the acidity of DES was not examined in this work, this way of discussing the results makes no sense. It should be corroborated with data of acidity from other sources. When authors comment the effect of molar ratio of HBA:HBD, it should be given the comparison. Here the effect of different HBA and HBD components of DES on the extraction efficiency can also be commented.

We appreciate the reviewer's attention to the flaws of our text. The stronger acidity of HBD is accompanied by higher desulfurisation efficiency. When DES is ChCl/p-TsOH (HBA:HBD = 1), the desulfurisation rate of the composite is 94.04%, and DES is ChCl/2p-TsOH, the desulfurisation rate is 95.8%. When HBD is SSA, the desulfurisation efficiency of ChCl/SSA composite is higher than that of TEAC/SSA.

3. Line 18: In the Effect of the oxidant section and further, mostly ChCl/p-TsOH and TEAC/SSA were used for all the experiments. It should be explained why these DESs were chosen.

Thanks for the reviewer's suggestion. ChCl/p-TsOH was chosen because it has the best desulfurisation efficiency for composite materials synthesized as a deep eutectic solvent; although TEAC/SSA has low desulfurisation efficiency for composite materials synthesized as a deep eutectic solvent, it will be more convincing to compare it with ChCl/p-TsOH in order to obtain accurate reaction conditions. Several other deep eutectic solvents with a large difference in desulfurisation efficiency from ChCl/p-TsOH can also be selected, of which we choose any one for comparison.

4. Line 32: 'hydrogen peroxide' is duplicate.

The reviewer has made a good point. We have already deleted the excess 'hydrogen peroxide'.

5. Figure 4 – what is TBHP? It is not mentioned anywhere in the text.

Thanks for the question. Actually TBHP is the abbreviation for tert-butanol hydrogen peroxide.

6. Figure 9 – why molar ratio of (DESs/CNTs):Oil:H₂O₂=1:5:0.5 was chosen?

Thanks for the question. In this experiment, 0.1g DESs/CNTs and 0.5g oil were selected for desulfurisation reaction. When exploring the influence of the amount of H₂O₂ on the desulfurisation efficiency, it was found that the desulfurisation efficiency of 0.05g H₂O₂ was the best, So (DESs/CNTs): Oil: H₂O₂ = 1: 5: 0.5 was chosen to study other factors.

7. The title of section 3.4. does not correspond with the text.

Thanks for the question. The content of section 3.4 is multiple removal and the recycling of composite materials. To realise deep desulfurisation of fuel oil, several desulfurisation processes were explored. A new composite material is added to the fuel oil after desulfurisation, the extraction process is performed under the same experimental conditions, and the fuel oil is reused. The regeneration process was performed by dissolving the composite materials in methyl tert-butyl ether which can wash off the excess fuel oil.

8. In section 3.6., line 56, '...intensity of DES/CNT is greater, while DES and DES/CNT are not.', the second 'DES/CNT' probably should be 'CNT'.

We appreciate the reviewers' attention to the flaws of our text. We have already changed DES/CNT to CNT.

9. Some relevant reports about using nanotubes for fuel desulfurization have been published, the authors are suggested to list their results and do some comparisons.

Thanks for the reviewer's suggestion. The adsorption desulfurisation efficiency of DBT by the CNTs was 60.88% in Sadare's group. The CNT/TiO₂ nanomaterials initially afforded approximately 45% adsorption removal of DBT, 55% BT, and more than 65% thiophene compounds from model fuels in Tawfik's group. The DBT oxidation catalyzed by CNTs@PDDA@Mo₁₆V₂ can reach to 99.4% in Gao's group. Novel carbon nanotube hybrids with good catalytic properties can be obtained by a facile and fast method for immobilization of WO₃/MoO₃ on the surface and into the

cavity of CNTs. The best oxidative desulfurization results (99.2%) have been obtained on the $2\text{WO}_3/\text{MoO}_3\text{-CNT}$ with the 16.5:8.5 % wt. of $\text{WO}_3:\text{MoO}_3$ in Afsharpour's group.

Reference section

1. References should be uniform as recommended by the journal.

For the next submission, the authors are suggested to prepare the manuscript exclusively according to the instructions of the journal (to correct the text – space, italic symbols etc., graphs) and to improve the English.

Thanks for the reviewer's suggestion. We have already written references according to the instructions of the journal (to correct the text – space, italic symbols etc., graphs).

Once again, Authors would like to express our appreciation to you and reviewers for suggestion how to improve our manuscript.

Best regards

Yue Liu, Jiaojiao Xue, Xin Zhou, Yingna Cui and Jingmei Yin

Appendix B

Dear Dr Laura Smith,

We thank “Royal Society Open Science” for giving us the opportunity to revise our manuscript. We appreciate Dr Laura Smith and reviewers very much for your comments and suggestions on our manuscript entitled " Deep desulfurisation performance of thiophene with deep eutectic solvents loaded carbon nanotubes composites "(ID: RSOS-201736).

We have studied reviewers' comments and have revised the manuscript according to your kind advices and reviewers' detailed suggestions. Enclosed please find responses to reviewers. The original manuscript was modified based on reviewers' suggestions. Below you will find reviewers' comments (in black) and our reply to each comment (in blue).

Reviewer: 2

Comments to the Author(s)

Most of the suggestions have been corrected, but the manuscript still needs minor revision. The authors are suggested to include in the manuscript their answers on Remark 6 from previous revision:

6. Figure 9 – why molar ratio of (DESs/CNTs):Oil:H₂O₂=1:5:0.5 was chosen?

Thanks for the question. In this experiment, 0.1g DESs/CNTs and 0.5g oil were selected for desulfurisation reaction. When exploring the influence of the amount of H₂O₂ on the desulfurisation efficiency, it was found that the desulfurisation efficiency of 0.05g H₂O₂ was the best, So (DESs/CNTs): Oil: H₂O₂ = 1: 5: 0.5 was chosen to study other factors.

We would like to thank the reviewer for his constructive and insightful criticism and suggestions. We have carefully discussed the issues raised by the reviewer and added this part in the manuscript.

Once again, Authors would like to express our appreciation to you and reviewers for suggestion how to improve our manuscript.

Best regards

Yue Liu, Jiaojiao Xue, Xin Zhou, Yingna Cui and Jingmei Yin